# The Role of Physical Exercise in Chronic Musculoskeletal Pain: Best Medicine—A Narrative Review

**DOI:** 10.3390/healthcare12020242

**Published:** 2024-01-18

**Authors:** Hortensia De la Corte-Rodriguez, Juan M. Roman-Belmonte, Cristina Resino-Luis, Jorge Madrid-Gonzalez, Emerito Carlos Rodriguez-Merchan

**Affiliations:** 1Department of Physical Medicine and Rehabilitation, La Paz University Hospital, 28046 Madrid, Spain; jmadridg@salud.madrid.org; 2IdiPAZ Institute for Health Research, 28046 Madrid, Spain; 3Department of Physical Medicine and Rehabilitation, Cruz Roja San José y Santa Adela University Hospital, 28003 Madrid, Spain; juanmaromanbelmonte@gmail.com (J.M.R.-B.); cristinaresino@salud.madrid.org (C.R.-L.); 4Medical School, Universidad Alfonso X El Sabio (UAX), 28691 Madrid, Spain; 5Department of Orthopedic Surgery, La Paz University Hospital, 28046 Madrid, Spain; ecrmerchan@hotmail.com; 6Osteoarticular Surgery Research, Hospital La Paz Institute for Health Research—IdiPAZ (La Paz University Hospital—Autonomous University of Madrid), 28046 Madrid, Spain

**Keywords:** musculoskeletal pain, chronic pain, musculoskeletal diseases, cost–benefit analysis, physical exercise

## Abstract

The aim of this paper is to provide a narrative review of the effects of physical exercise in the treatment of chronic musculoskeletal pain. Physical inactivity and sedentary behavior are associated with chronic musculoskeletal pain and can aggravate it. For the management of musculoskeletal pain, physical exercise is an effective, cheap, and safe therapeutic option, given that it does not produce the adverse effects of pharmacological treatments or invasive techniques. In addition to its analgesic capacity, physical exercise has an effect on other pain-related areas, such as sleep quality, activities of daily living, quality of life, physical function, and emotion. In general, even during periods of acute pain, maintaining a minimum level of physical activity can be beneficial. Programs that combine several of the various exercise modalities (aerobic, strengthening, flexibility, and balance), known as multicomponent exercise, can be more effective and better adapted to clinical conditions. For chronic pain, the greatest benefits typically occur with programs performed at light-to-moderate intensity and at a frequency of two to three times per week for at least 4 weeks. Exercise programs should be tailored to the specific needs of each patient based on clinical guidelines and World Health Organization recommendations. Given that adherence to physical exercise is a major problem, it is important to empower patients and facilitate lifestyle change. There is strong evidence of the analgesic effect of physical exercise in multiple pathologies, such as in osteoarthritis, chronic low back pain, rheumatoid arthritis, and fibromyalgia.

## 1. Introduction

Musculoskeletal (MSK) disorders are responsible for more than one-third of years lived with pain and represent one of the most important causes of disability in the world [1].

The International Association for the Study of Pain defines pain as “an unpleasant sensory and emotional experience associated with, or resembling that associated with, actual or potential tissue damage” [2]. Up to 80% of chronic pain is MSK in origin [3].

Chronic pain is also associated with premature aging, given that telomere length (a biomarker related to age-related pathology) has been found to be shorter in the chronic pain population [4]. This is of particular concern, because aging is associated with the risk of developing severe disease, especially after the age of 40 [5]. This relationship between pain and telomere length is only found in those who do not exercise regularly [6].

Musculoskeletal pain is a major drain on health-care resources, being a leading cause of incapacity for work and sick leave [7]. Of patients with chronic pain, 51.5% experience disability. The results of this study revealed that no habit of walking or working out were more likely to be associated with disability [8]. Globally, low back pain is believed to be the leading cause of years lived with disability [7]. In the working population, around 2% of gross domestic product is lost [9]. Physical exercise is a type of structured, planned, repetitive physical activity that promotes the maintenance or development of physical fitness [10]. Performed regularly, physical exercise is considered a non-pharmacological intervention with many beneficial health effects [11]. For the management of MSK pain, physical exercise is an effective, cheap, and safe therapeutic option, because it does not produce the adverse effects of pharmacological treatments or invasive techniques [12]. In addition to its analgesic capacity, physical exercise has an effect on other pain-related areas, such as sleep quality, activities of daily living, quality of life, physical function, and emotional impact [13].

The aim of this paper is to provide a narrative review of the effects of physical exercise in the treatment of chronic musculoskeletal pain.

## 2. Materials and Methods

With the goal of creating a narrative overview of the most recent research on this subject, a literature search was undertaken on 15 October 2023. The search terms “exercise musculoskeletal chronic pain” were used in the PubMed, Cochrane, Embase, CINAHL, and PEDro databases. The inclusion criteria followed were clinical studies conducted in adults (>18 years) with common chronic musculoskeletal pathology: back pain, cervical pain, osteoarthritis, fibromyalgia or multisite pain (more than one location). Articles in English with no date limit were included. Laboratory and animal studies describing pathophysiological mechanisms of exercise-mediated analgesia were also included. We selected studies that included physical exercise as an intervention and that included pain, quality of life or work related outcomes among the outcomes analyzed. Within the exclusion criteria, editorials, letters, or commentaries were not considered in the present review, nor were articles whose language was not English. We also did not include studies that assessed chronic musculoskeletal pain of malignant or traumatic origin or in which other less frequent or different musculoskeletal pathologies coexisted. When multiple references on the same topic were found, a bibliographic selection was conducted based on the robustness of the evidence. The included articles were as follows: (1) higher quality of evidence, such as meta-analyses, systematic reviews, and randomized clinical trials; (2) recent publications that offered new information not found in other literature works; (3) articles that described the physical exercise protocol for pain and that could be used to produce consistent clinical recommendations. In total, 2860 articles were located: 1465 in PubMed, 931 in Embase, 60 in CINAHL, 394 in PEDro, and 10 in the Cochrane Database. Overall, 198 records were chosen after duplicate articles were found and 134 publications were ultimately included in this review out of the 152 that were considered interesting and closely linked to the article’s topic (Figure 1). No specific software or digital bibliographic reference manager was used. A database was created with the papers from the different search platforms, and a manual comparison was made to detect duplicates. Subsequently, the data were collected for review. Data extraction, searches and duplicate detection were carried out by colleagues in the study. Cases of doubt were resolved by discussion among the different authors. Eligible articles after eliminating duplicates (1704) were screened by title and abstract. From these, 209 articles were selected for full-text review: 152 were selected for eligibility and finally 134 were included in this study.

## 3. Results and Discussion

### 3.1. Sedentary Lifestyles and Chronic Pain

“Sedentary behavior” refers to activities that do not increase energy expenditure substantially above resting level. It includes activities such as sleeping, sitting, lying down, watching television, and other forms of screen-based entertainment [14,15,16].

Lack of physical exercise implies a decrease in the mechanical stress on the muscle–tendon structures, which can lead to deterioration of these tissues [17]. In addition, the lack of load on the joints is associated with a decrease in bone mineral density [18]. Physical inactivity has also been found to lead to rapid loss of muscle mass and to degenerative changes in the nervous system [19].

The World Health Organization (WHO) recommends that adults engage in at least 150 min of moderate exercise or 75 min of vigorous exercise each week [20]. However, it has been reported that one in four adults do not meet the minimum physical exercise recommendations proposed by the WHO in 2020 [21].

Adults with more sedentary lifestyles are more likely to experience some form of pain [22]. The onset and development of chronic MSK pain is also associated with the adoption of sedentary behaviors [23].

One of the major changes occurring in today’s work environment is the multitude of sedentary jobs that encourage physical inactivity [24]. This physical inactivity can lead to the development of chronic disease in up to 15% of men and 20% of women [25], including obesity, type 2 diabetes mellitus, heart disease, and other chronic diseases with high morbidity and mortality [26].

We know that a sedentary lifestyle promotes weight gain. Obesity is a comorbidity frequently associated with chronic pain [27] and increased opioid consumption [28].

Pain has also been reported to be a barrier to physical activity in patients with obesity [29]. The relationship between obesity and pain is complex and multidirectional, with multiple physiological mechanisms involved at the level of neurological and metabolic modulation [30].

Other forebrain phenomena, such as cognition, attention, and emotions, have been shown to modulate the clinical experience of pain and could contribute to the mechanism of central sensitization [31]. In this sense, psychological stressors are powerful pain triggers, and measures should be taken to prevent their occurrence [32].

Patients with severe mental illness have a high pain incidence, which is associated with decreased quality of life [33]. These patients are also at increased risk of developing cardiovascular complications and obesity [34], in many cases due to physical inactivity [35].

### 3.2. Physical Exercise as Analgesic Therapy

The exact relationship between physical exercise and pain is not yet fully understood [36]. However, it appears clear that sustained physical exercise is beneficial in terms of pain [37]. Physical exercise has a preventive role in the development of chronic pain [38]. One study found that people with an active lifestyle had a lower sensitivity to thermal stimuli, which could imply that they feel less pain [39]. When a healthy person performs an acute bout of exercise, a period of hypoalgesia occurs in which there is a decrease in sensitivity to painful stimuli of variable duration [40]. In athletes, a decrease in pain sensitivity has been observed after 120 min of aerobic exercise [41].

The role of physical exercise in preventing the onset of pain might be important. A systematic review of 21 randomized clinical trials including 30,850 participants found a low level of evidence that an isolated exercise program can prevent the occurrence of chronic low back pain episodes and moderate evidence when exercise is combined with education [42]. Physical exercise also appears to improve pain associated with a large number of diseases [13]. For this, exercise should be part of a multimodal pain therapy protocol, which should include other physical and psychological aspects, emphasize elements such as self-efficacy and empowerment, and respect the patient’s preferences and beliefs [3].

Within multimodal therapy, one of the most frequently recommended treatments is patient education, including advice on activity, healthy lifestyles, and weight loss [43], which can help the patient to better understand pain mechanisms and how to cope, and can reduce kinesiophobia. It is important to encourage treatments in which the patient plays an active role so as to promote self-efficacy, which is associated with improved chronic pain outcomes and quality of life [44,45].

A review of 21 Cochrane reviews (381 studies, 37,143 participants) reported that physical exercise might improve pain and physical function in conditions as varied as rheumatoid arthritis, osteoarthritis, patellofemoral pain, fibromyalgia, low back pain, chronic neck pain, intermittent claudication, dysmenorrhea, post-polio syndrome, and spinal cord injury [46].

Recommendations on specific pathologies are discussed below.

### 3.3. Physiology of Exercise-Mediated Analgesia

The experience of pain is a complex phenomenon in which many other processes besides tissue injury play an important role [47]. The analgesia achieved by physical exercise appears to be due to the coordinated and synergistic action of two systems—opioid and non-opioid [48].

In the interstitial space are located free nociceptive nerve endings whose excitation is transmitted afferently to produce pain perception. These nociceptors respond to the chemicals produced by MSK injury. However, the pain’s intensity depends not only on the concentration of these substances but also on the excitation threshold of the nociceptors. Thus, in peripheral sensitization, a non-painful stimulus can activate these nociceptors, causing pain to be perceived. This process repeated over time can lead to central sensitization [49], which is directly related to chronic pain. This pain can be nociceptive, neuropathic, or non-nociplastic, and most frequently mixed. Chronic pain in several musculoskeletal conditions is thought to be the result of abnormal central pain processing rather than tissue injury [47]. Aerobic physical exercise might decrease both central and peripheral pain sensitivity [50].

Chronic MSK pain causes alterations in brain structure and function [51], reducing the volume and density of gray and white matter [52]. It can also lead to alterations in the connections of various brain regions [53]. Within the brain structures, the insula is a key region in the processing of pain patterns, and its activation levels are related to pain intensity. A short- or long-term physical exercise program can produce changes at this level [54]. Descending and endogenous inhibitory pathways also play an important role and are activated during exercise, leading to a decrease in pain perception after exercise [55].

The analgesia achieved by physical exercise is due to the coordinated effect it has on the central nervous system, the hormonal system, and at the level of inflammatory markers. The opioid system together with the endocannabinoid system is responsible for most of the analgesia achieved by physical exercise [48]. At the hormonal level, progesterone has a known analgesic effect, whereas the other hormonal analgesia pathways are more debated, although they might improve pain through their anti-inflammatory and reparative effect [48].

#### 3.3.1. Effects of Physical Exercise on the Central Nervous System (Neurochemical)

##### Endogenous Opioids

The role of endorphins in the central effect of exercise has been discussed due to their difficulty in crossing the blood–brain barrier [56]. Aerobic training increases levels of β-endorphin and met-enkephalin in the hypothalamus, periaqueductal gray, and rostral ventromedial medulla [57,58], which increase as exercise intensity increases [59]. If training is maintained for more than 9 weeks or 45 sessions, downregulation of the opioid system occurs, and a significant reduction in mu-opioid receptors has been found [60].

##### Endocannabinoid System

The endocannabinoid system is composed of cannabinoid receptors (CB1, CB2), their ligands (n-arachidonoylethanolamine [AEA] or anandamide and 2-arachidonoylglycerol [2-AG]), and proteins involved in their metabolism. Endocannabinoid CB1 and CB2 receptors located in the rostral ventromedial medulla and periaqueductal gray dorsal horn [60] produce analgesia by binding to AEA and 2-AG, two neurotransmitters whose levels increase with exercise and that modulate synaptic activity and neuronal plasticity [61]. A resistance exercise program can increase CB1 receptor expression in brain tissue and in periaqueductal regions [62].

##### Serotonergic System

Serotonergic system-mediated analgesia is modulated through 5-HT receptors and is regulated by the serotonin transporter [63]. Exercise increases the concentration of 5-HT receptors in the lumbar spinal cord, brainstem, and parieto-occipital cortex [64]. Regular exercise activates opioid receptors in the periaqueductal gray, which in turn modulate serotonin transporter activity [57]. It also increases the concentration of serotonin in the brain, which plays a role at the emotional and memory levels [65].

##### N-Methyl D-Aspartate Receptor Alteration and the Noradrenergic System

The phosphorylation of the N-methyl D-aspartate receptor NR1 subunit in the rostral ventromedial medulla produces hyperalgesia, and its inhibition produces analgesia [66]. Physical exercise activates the noradrenergic system, which produces catecholamines that modulate pain by binding to their receptors (α1, α2, β2 adrenergic) located in the periaqueductal gray, dorsal raphe, and spinal cord dorsal root ganglion [67].

The central nervous system’s analgesic mechanisms during exercise are described in Table 1 [48,68,69,70].

#### 3.3.2. Effect of Physical Exercise at the Hormonal Level (Immunomodulatory)

Progesterone has an anti-inflammatory effect by regulating prostaglandins and leukotrienes and has an analgesic effect by modulating pain circuits [71]. Physical exercise produces an increase in progesterone concentration, depending on the intensity of training, reaching a maximum concentration at exhaustion, especially in untrained individuals [72]. The anti-inflammatory effect activated by arachidonic acid, prostaglandins, and inflammatory markers might also have an indirect effect on pain [73,74,75].

Physical exercise increases the synthesis of prostaglandins, especially PGE2 and PGF2-alpha in skeletal muscle, which regulates protein synthesis in muscle [74].

Arachidonic acid has an anti-inflammatory and pro-fibrinolytic effect [73]. Arachidonic acid is metabolized to prostaglandins and leukotrienes and through the cytochrome P450 pathway gives rise to other metabolites such as epoxyeicosatrienoic acid, which in turn is transformed into dihydroxyeicosatrienoic acid [76].

Physical exercise decreases the levels of pro-inflammatory cytokines (TNF-α, IL-6, IL-1β) and activates macrophages that release anti-inflammatory cytokines (IL-10) [75]. In addition, physical exercise appears to modulate C-reactive protein [77].

### 3.4. Prescription of Analgesic Physical Exercise

As previously mentioned, no single therapy achieves significant pain relief; thus, recommendations tend to be for multimodal treatments that maximize beneficial effects and minimize adverse effects [78]. There are various types of physical exercise, including aerobic, strength training, flexibility, and balance. Programs that combine several of these modalities (multicomponent exercise) are typically more effective and can be better adapted to clinical conditions. Before prescribing exercise, it is important to assess the patient to determine those biomechanical aspects (lack of strength, stiffness, difficulty with endurance or motor control) that will allow for individualized prescription of a physical exercise program aimed at compensating for these deficits [79].

Given that not all physical exercise programs have the same effect, it is important to derive a correct prescription. Within the physical exercise prescription, type, intensity, frequency, time, and duration should be considered (Table 2).

Most physical exercise protocols do not use the same interventions (frequency, duration, intensity, modality) or assess the same pathologies in a homogeneous way. There is a lack of unified protocols to compare different physical exercise interventions with each other. Furthermore, there are potential biases in the studies analyzed (different baselines in the participants, treatment not clearly specified) that should be taken into account in order to critically evaluate these recommendations. Therefore, it is difficult to establish solid recommendations due to the heterogeneity of studies on physical exercise and chronic MSK pain.

#### 3.4.1. Type of Exercise

Regarding the type of exercise, the hypoalgesic effect occurs with both aerobic and strengthening exercises. However, stretching and proprioceptive exercises have been shown to be less effective for pain [80]. Aerobic exercise involves the continuous movement of all the major muscle groups acting in a coordinated manner to produce a cardio-metabolic stimulus [81]. Performing aerobic exercise at a low-to-moderate intensity of 50–60% of the maximum heart rate has an analgesic effect in chronic pain pathology [82].

Strengthening exercises involve the contraction of muscles to overcome resistance. They are effective in chronic pain, being well tolerated and achieving better results when land-based [13] exercises.

Stretching exercises aim to improve joint range of motion and muscle flexibility. Proprioceptive exercises promote neuromuscular training to improve posture, balance, and coordination. Although they have little effect on pain when performed on their own, they might have more effect on an emotional level; thus, it is recommended to include them within training programs that include other exercises [83].

#### 3.4.2. Exercise Intensity

When calculating exercise prescription intensity, vigorous aerobic exercise (70% heart rate recovery [HRR], 3 sessions, 25 min each session) has been reported to have a higher hypoalgesic effect, producing a greater decrease in pain response to thermal stimulation and increase in pain threshold than moderate physical exercise (50% HRR, 3 sessions, 25 min each session) [84]. Other studies, however, have found that aerobic exercise (treadmill running, 35 min) at low (40% HRR) or moderate (55% HRR) intensity produces a greater primary and delayed (persisting more than 24 h) hypoalgesic response than intense exercise (70% HRR) [85]. Other studies have found no difference in hypoalgesia with aerobic exercise as a function of exercise intensity [86,87].

In strengthening exercises, performing isometric contractions with loads of only 10–30% is sufficient to achieve a hypoalgesic effect, especially when the duration of the contraction is prolonged until failure [88]. A 3 min program of wall squat exercise also produces a hypoalgesic effect [89].

#### 3.4.3. Exercise Frequency

There is no solid evidence to recommend specific parameters in terms of frequency. In knee osteoarthritis, it has been reported that physical exercise programs of 3 days a week are more effective in improving pain and function than less frequent training [90].

**Table 2 healthcare-12-00242-t002:** Parameters to be taken into account when prescribing physical exercise [91].

Type	Aerobic, strengthening, stretching or proprioceptive (less commonly used for analgesia)
Intensity	Degree of exertion achieved during exercise
Frequency	Sessions of exercise per week
Time	Minutes of exercise per week
Duration	Number of weeks the exercise is carried out

#### 3.4.4. Timing and Duration of the Exercise

A more recent meta-analysis studying the effect of exercise time on chronic pain reported that to achieve an optimal analgesic effect, physical exercise programs of no more than 120 min per week should be performed [91]. Another study compared three exercise programs using a bicycle ergometer at 75% VO_2_max for 30, 45, and 60 min without finding differences among the groups (in fact, no hypoalgesic effect was reported in this study) [92].

Regarding the duration of exercise in chronic pain, one study reported that to achieve an optimal analgesic effect, physical exercise programs should be performed for a minimum of 7 weeks and a maximum of 15 weeks. After this period, the analgesic effect associated with exercise begins to decline [91]. The analgesic response to physical exercise is variable in patients with chronic pain [46]. Given that the analgesic response depends on the individual patient and the intensity and modality of exercise, it is recommended that the prescription always be individualized [93] and prescribed by professionals with expertise in MSK pathology, exercise, and pain.

Table 3 summarizes the analgesic effect according to the characteristics of the prescribed exercise program.

### 3.5. Pain Tolerance during Exercise

The ideal dose of physical exercise needed to improve pain, depending on the patient and the pathology, is not yet known. The perception of pain during exercise is important for safety and exercise adherence.

There is a risk that patients will discontinue exercise if they perform insufficient exercise that does not provide benefit or if they perform excessive exercise that causes pain [85]. Therefore, exercise programs should be personalized to minimize adherence problems that can result in failure to achieve the desired long-term benefits [94].

However, it appears that the analgesic effect achieved with an exercise program depends more on the exercise’s intensity and volume than on its frequency [95]. If moderate-intensity physical exercise is performed either before or after a pain episode, greater hypoalgesia will be achieved. Conversely, if the intensity of that exercise reaches the point of exhaustion, the effect will be reduced and pain will increase [96]. In chronic MSK pain, however, painful exercises have been reported to offer a small but significant benefit compared with pain-free exercises in the short term [97]. The recommendation for musculoskeletal rehabilitation (e.g., for osteoarthritis) is to use safety thresholds based on the visual analogue scale (VAS, 0–10). Pain perception up to 2/10 is considered “safe”, pain up to a level of 5/10 is considered “acceptable”, and pain above 5/10 is considered “high risk” [98]. Therefore, the most beneficial program would start with a low intensity and work up to a higher intensity as tolerated by the patient [91].

Exercises performed in water are frequently recommended for the treatment of chronic pain because the body weighs less in this environment and because of its thermal capacity [99]. A program involving swimming for 50 to 90 min 5 days a week or treadmill walking for 10 to 60 min 3 to 7 days a week reduces mechanical pain and improves thermal sensitivity [100,101].

On the other hand, chronic MSK pain might not correlate with the level of tissue damage [102]. Catastrophizing and fear-avoidance behaviors can play an important role in pain response and exercise tolerability [103]. Therefore, associating behavioral coping strategies with pain is essential to address tolerability to exercise modalities. If patients find the prescribed exercise not acceptable or safe, clinical benefit or adherence to strength exercise could be limited.

Conversely, adequate physical exercise can lead to an improvement in patients’ mood, which can improve adherence to physical treatment programs and facilitate lifestyle change [104]. Given that adherence to physical exercise is a major problem, it is important to empower patients and facilitate lifestyle change. Other variables, such as exercise equipment, supervision during protocols, and therapist experience, are also important for adherence to exercise programs [98].

### 3.6. Exercises and Analgesia in Various Musculoskeletal Pathologies

For most pathologies that cause chronic MSK pain, physical exercise is often recommended to improve strength, flexibility, physical fitness, and general health [46]. It is important that the patient is actively involved in the prescription of their training program and to express their preferences, expectations, and barriers [105].

#### 3.6.1. Chronic Mechanical Low Back Pain

Contrary to common belief, recreational sport and physical activity are not associated with an increase in low back pain occurrence [106]. Physical exercise is recommended in the treatment of chronic mechanical low back pain, either with or without radiculopathy. Physical exercise can produce a small improvement in the short term (up to 6 months) [107] and moderate improvement in the long term (more than 12 months) [108].

An aquatic exercise program (12 weeks’ duration, 2 sessions per week of 60 min each, supervised by physiotherapists) can produce benefits for up to 12 months in pain intensity, sleep quality, quality of life, fear avoidance, and kinesiophobia [109].

Although swimming also appears to be beneficial, the butterfly stroke should be avoided, given that it could lead to activation of the directing muscles of the back and hyperlordosis, which can promote recurrence of low back pain or even spondylolysis [110].

Yoga involves the performance of a series of postures, as well as breathing and meditation techniques based on a Hindu philosophy. It appears that the performance of yoga can produce an improvement in low back pain and function compared with usual care, education, or other exercises [111,112].

Tai chi is a set of techniques based on Chinese martial arts that are based on slow and fluid movements, which has been found to have an analgesic effect on mechanical lumbago, fibromyalgia, and osteoarthritis [113].

Walking, one of the most common forms of physical activity, appears to be effective in improving low back pain in both the acute and chronic phases [114,115]. When walking slowly, however, the amplitude of the swing decreases, which can increase pressure on the lumbopelvic area; thus, brisk walking is recommended [116]. Regarding running, moderate running does not appear to worsen low back pain, provided it is performed progressively and that appropriate footwear is worn [117]. Intensive running is not recommended because it can increase the incidence of low back pain [118].

Cycling can be beneficial for patients with low back pain because it involves significant aerobic activity; however, it could worsen pain in those who are not sufficiently fit [119]. The position in which the cyclist sits promotes kyphosis and could make cycling poorly tolerated. In this sense, adjusting the bicycle (changing the saddle angle) can reduce the intensity and frequency of the onset of low back pain by approximately 70% [120].

#### 3.6.2. Chronic Cervical Pain

General physical exercise does not appear to be of great benefit in neck pain; however, specific physical exercise to strengthen the neck, shoulders, and upper back does appear to reduce chronic neck pain [121].

Physical exercise appears to have more analgesic effect than other techniques such as massage. A clinical trial compared a cervical and scapular muscle strengthening exercise program with massage, finding that patients performing resistance exercise improved pain, cervical ROM, upper trapezius tone, disability level, and QOL compared to the massage group [122].

Among the various types of exercise used to treat neck pain, the most effective in improving pain intensity and disability are strengthening exercises, motor control exercises, and yoga/tai chi/Pilates [123].

#### 3.6.3. Fibromyalgia

People with fibromyalgia often catastrophize, which implies a state of pessimism, helplessness, and rumination about pain-related symptoms [124]. Given that catastrophizing has an inverse relationship with muscle endurance [125], it is important to avoid pain or fatigue during physical exercise, because it has been associated with poor adherence to physical exercise programs when it occurs [104].

Strengthening exercises, aerobic exercises, or a combination of these can produce a small improvement in pain and function in patients with fibromyalgia [126,127]. It has also been reported that a low-impact physical exercise protocol combining resistance and coordination exercises might be effective in improving pain perception, psychological aspects (catastrophizing, anxiety, depression), quality of life, and physical fitness [104].

In one study, it was found that aquatic training improved pain threshold, visual analogue scale pain (*p* = 0.01), well-being and lower Fibromyalgia Impact Questionnaire score. However, these benefits were lost after a detraining period of 16 weeks [128]. A systematic review analyzing 16 studies (881 patients) found low-to-moderate evidence of a beneficial effect of aquatic exercise on wellness, symptoms, and fitness in adults with fibromyalgia [127] A subsequent meta-analysis analyzed 22 articles (1722 patients) and found that aquatic exercise produced an improvement in pain, sleep quality and quality of life in fibromyalgia patients [129].

#### 3.6.4. Hip and Knee Osteoarthritis

In osteoarthritis of the hip, strengthening, stretching, endurance, and neuromuscular control exercises are beneficial. They can produce short-term pain improvement, although not in the medium term [100].

In osteoarthritis of the knee, stretching exercises, gait training, and walking are beneficial. These exercises can produce a small-to-moderate improvement in pain in the short to medium term, but not in the long term [130]. Within strengthening exercises, both concentric and eccentric strengthening exercises appear to be effective in improving pain, although concentric exercises appear to decrease ambulatory pain and pain upon walking cessation more [131].

Aquatic exercise can produce a small short-term improvement in pain, function, and quality of life in both hip and knee osteoarthritis [132].

A Cochrane review on non-pharmacological interventions in rheumatoid arthritis found moderate evidence for the effect of physical exercise in improving fatigue, a condition that for these patients can be as limiting as pain [133]. Regarding pain, the performance of an exercise program (twice per week, moderate loads of 50–70% of 1 repetition maximum, 2 sets per exercise, 8–12 repetitions per set) achieved an analgesic effect at 12 months (0.31 mean on the VAS) and at 24 months (1.1 mean on the VAS) compared with a control group [134].

## 4. Limitations of the Study

In this narrative review, the papers that were included were considered most significant in relation to the article’s title, giving priority to the research that provided the greatest amount of evidence.

Pain-mediated exercise has numerous mechanisms of action, but neither the entire process nor the best application practices are fully understood at this time. Actually, there is a lot of variation in exercise regimens, which makes it challenging to compare studies.

In the future, more research on the effectiveness of exercise as pain management for MSK disorders would be ideal. To use the approach as effectively as possible, further information on the best treatment plans would also be necessary. Exercise has a wide range of mechanisms of action, and thus it is likely that new treatments for MSK pain will keep developing.

## 5. Conclusions

Sedentary behavior and physical inactivity are associated with chronic musculoskeletal pain and can aggravate it. For the management of this pain, physical exercise is an effective, inexpensive, and safe therapeutic option, because it does not produce the adverse effects of pharmacological treatments or invasive techniques.

In addition, its analgesic capacity has an effect on other pain-related areas, such as sleep quality, activities of daily living, quality of life, physical function, and emotional impact. In general, even during acute periods, maintaining a minimum level of physical activity can be beneficial.

Combining various types of exercise modalities (aerobic, strengthening, flexibility, and balance), known as multicomponent exercise, is often more effective and can be better adapted to the individual’s clinical conditions. For chronic pain, aerobic and resistance training programs appear to provide the greatest benefit at a moderate intensity and duration of 60–120 min for 7–15 weeks. Exercise programs should be tailored to the specific needs of each patient, based on patient characteristics and clinical guideline recommendations.

There is solid evidence of the analgesic effect of physical exercise in many pathologies, such as chronic low back pain, neck pain, osteoarthritis, fibromyalgia, and rheumatoid arthritis. However, the physical exercise protocols used as an intervention are often not well defined or are very heterogeneous, as are the outcome measures used. Therefore, it would be desirable that more studies with an adequate methodology be carried out to assess the role of physical exercise in the treatment of chronic pain, its mechanism of action and its short- and long-term effects.

## Figures and Tables

**Figure 1 healthcare-12-00242-f001:**
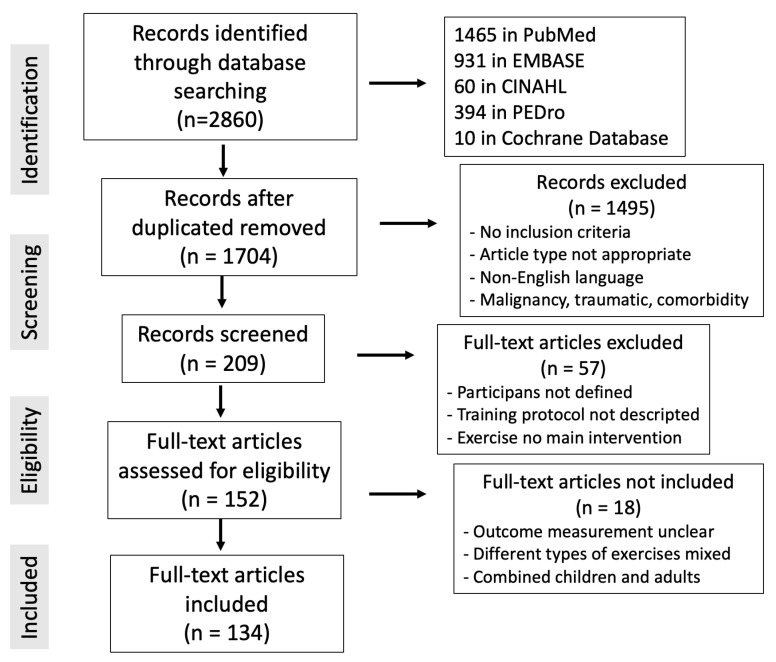
PRISMA flowchart of the literature review for this article and the terms “exercise musculoskeletal chronic pain”.

**Table 1 healthcare-12-00242-t001:** Central nervous system analgesic mechanisms in exercise [48,68,69,70].

	Brain Area	Neurotransmitters Involved	Receptors	Exercise Program	Intensity	Effect
Endogenous opioids	Hypothalamus, periaqueductal gray, rostral ventromedial medulla	β-endorphin, met-enkephalin	Mu-opioid	Aerobic training performed for 5–8 weeks	85% of maximum heart rate or 80% of VO_2_max	Training >9 weeks or 45 sessions produces downregulation
Endocannabinoid system	Rostral ventromedial medulla, periaqueductal gray dorsal horn	Anandamide, 2-arachidonoylglycerol	Endocannabinoid CB1 and CB2	Aerobics and resistance (running, followed by cycling)	70–85% of maximum heart rate or 25% maximum isometric contraction	“Runner’s high”, anxiolytic, sedative, and euphoriant
Serotonergic system	Brain stem, lumber spinal cord, and parieto-occipital cortex	Serotonin transporter (SERT)	5-HT	1 session of 60 min swimming or swimming 30 min/day, 6 days per week; run on a treadmill 30 min, 5 days per week, 2 weeks	Low intensity (75% maximal blood lactate steady state)	Emotional regulation and facilitated memory function in the hippocampus
NMDA receptor alteration	Rostral ventromedial medulla	Phosphorylation of N-methyl D-aspartate (NMDA)	Receptor NR1 subunit (p-NR1)	Exercise 1 h for 1 week	75–85% HRmax	Modulates nervous system’s response to pain stimuli and injury
Nor-adrenergic system	Periaqueductal gray, dorsal raphe, and spinal cord dorsal root ganglion	Catecholamines	α1, α2, β2 adrenergic	1 or 2 h treadmill running (A), 20 min of stationary bicycle (B)	Moderate (25 m/min with a 3% slope) (A), intense (Borg 15) (B)	Improves cognitive performance, modulates thermoregulation during exercise

**Table 3 healthcare-12-00242-t003:** Analgesic effect according to the characteristics of the exercise program performed: parameters to be taken into account when prescribing physical exercise.

Type	Aerobic	Strengthening	Stretching
Frequency	40–59 min	60–120 min	>120 min
Duration	4–6 weeks	7–15 weeks	>15 weeks
Intensity (HRR)	25–40% HRR	40–55% HRR	55–70% HRR
Intensity (load)	10–30% 1RM	30–50% 1RM	50–85% 1RM
Pain during exercise	0–2/10 VAS	>2–5/10 VAS	>5–10/10 VAS
Modality	Aquatic	Land-based	Nature

Green: most effective; orange: moderately effective; red: least effective. HRR: heart rate recovery; RM: repetition maximum; VAS: visual analogue scale.

## Data Availability

No new data were created or analyzed in this study. Data sharing is not applicable to this article.

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
