# Peer review of "The Role of Physical Exercise in Chronic Musculoskeletal Pain: Best Medicine—A Narrative Review"

_healthcare, 2024, doi:10.3390/healthcare12020242_

Round 1
Reviewer 1 Report
Comments and Suggestions for Authors
Comments and Suggestions for Authors
Thank you for the opportunity to examine this interesting work that review the literature on the effect of physical exercise on musculoskeletal pain.
I have some comments that I consider necessary to improve the presentation of the study.
TITLE
It is suggested to add the type of study "narrative review" to the title.
AUTHORS
Check this address: (4) Alfonso X el Sabio Universit.
KEYWORD
Instead of “Musculoskeletal”, my suggestion would be to include two meSH terms: “Musculoskeletal Pain” and “Chronic Pain”. The descriptor “Musculoskeletal Diseases” could also be added.
“Benefits” is a very non-specific term that could perhaps be replaced by the descriptor "Cost-Benefit Analysis".
1. INTRODUCTION
It is very brief.
LINES 49-53: It is recommended to cite sources from which this information is extracted.
2. MATERIALS AND METHODS
- This section must be developed with a greater level of detail so that the methodology would be reproducible in other investigations.
- Specify the search terms and search strategies used in each of the databases.
- Describe the selection criteria in more detail. As expressed, they are very generic. Both the inclusion and exclusion criteria should be described in more detail. Dates, language, level of evidence are not mentioned...
- To make the selection, have you reviewed the title, the title and abstract, or the full text? Has it been done by a single reviewer or by peers?
- LINE 68: develop abbreviation MSK.
- Has any bibliographic manager been used, any software to manage duplicate documents, to collect data, etc...?
- LINE 61 and LINE 68: There are data that are incoherent: “literature available on the topic (2860)” and “3508 articles in total were located”.
- Figure 1. PRISMA Flowchart: “3508 articles in total were located”. Review these data, they do not correspond to the Flowchart. It is recommended to represent the number of results obtained in each database in the Flowchart. Likewise, it is recommended to raise the two cells on the right of the Flowchart one level for a better interpretation of the figure. Provide reasons on the number of documents excluded.
3. RESULTS AND DISCUSSION
-Table 2. Indicate the bibliography from which the Parameters and their definition have been extracted.
-LINE 259: Add the quote from the meta-analysis studying.
-3.6.3. Fibromyalgia: It is suggested to add evidence on the beneficial effects of Aquatic exercise in Fibromyalgia.
4. REFERENCES
-Review the format of this section.
-The title of the journals must appear in abbreviated and italic format and the year of publication in bold.
-Check the number of authors in all references. Normally, after the 6th author we put et al. REF. No. 14, 15. 18,19, 29,30, 36, 37, 48, 63…
-REF. No. 16 review author.
-REF. No. 36, review it.
-It is recommended to add DOI to references that have it.
Author Response
Dear Reviewer 1
Thank you very much for all your suggestions and comments which will surely improve the quality of the article. We have answered your questions, you can find the answers below. In addition, we have added new sentences, paragraphs and references in the new version of the manuscript which you can find in red.
Warm regards,
Authors

Reviewer 2 Report
Comments and Suggestions for Authors
This is an interesting and well written manuscript. However, I have some comments and suggestions which may improve the quality pf this paper:
- Introduction section
- The introduction could provide more background on the prevalence and impact of musculoskeletal pain to emphasize why this is an important health issue to study. Adding some key statistics on the societal costs and disability associated with musculoskeletal pain would strengthen the rationale.
- More clarity is needed on the scope - currently it focuses on both acute and chronic musculoskeletal pain but does not define or differentiate these. It would be helpful to specify if the focus is on one or both pain types.
- Materials and Methods
- The search strategy and selection criteria for studies included in the narrative review could be described in more detail and systematically. Providing the specific search terms, databases, inclusion/exclusion criteria would improve reproducibility.
- There is no critical appraisal of the quality of studies included. Assessing risk of bias or quality of evidence would strengthen the methodology.
- Results/Discussion
- The results rely heavily on citing previous review articles. Including some discussion of original studies/trials on key points could enhance the review.
- More detail is needed on the underlying mechanisms described e.g. referring to specific receptor pathways. The physiology section is vague in places.
- In exercise prescription recommendations, factors like optimal duration and frequency are unclear. Providing summary boxes/tables to clearly lay out evidence-based recommendations for practice would be helpful.
- There is little critical analysis of the evidence base throughout. The limitations around heterogeneity of protocols and pain models should be addressed.
- Conclusions
- The conclusions present a rather uncritical summary and do not highlight limitations around incomplete evidence or need for further research. This could be expanded on.
Author Response
Dear Reviewer 2
Thank you very much for all your suggestions and comments which will surely improve the quality of the article. We have answered your questions, you can find the answers below. In addition, we have added new sentences, paragraphs and references in the new version of the manuscript which you can find in blue.
Warm regards,
Authors

Round 2
Reviewer 1 Report
Comments and Suggestions for Authors
The authors revised the paper in response to the earlier comments and suggestions. The manuscript is likely publishable pending an additional revision.
(4) Alfonso X El Sabio University.
Abstract: The Aim should be placed at the beginning, not the end of the Abstract.
Keywords: Musculoskeletal Pain, Chronic Pain, Musculoskeletal Diseases, Cost-Benefit Analysis are appropriate, and please add Physical exercise.
Materials and Methods
Only one search term "exercise musculoskeletal pain" was used?. Specify all search terms.
LINES 89-90: Only the data extraction was carried out by pairs? And the rest of the process (searches, detect duplicates, etc.) how was it done?
Figure 1. PRISMA flowchart: the reasons why 152 studies assessed for eligibility become 134 articles are not argued. Add this information please.
Author Response
Dear Reviewer 1
Thank you very much for this new revision. You can find responses in the file attached and new information in green in the new version of the manuscript.
Warm regards,
Authors

Reviewer 2 Report
Comments and Suggestions for Authors
The authors responded to my comments and suggestions very well. Thank you.
Author Response
Dear Reviewer 2
Thank you very much for this new revision and your words.
Warm regards,
Authors